# BEYOND SINGLE-AXIS FAIRNESS: LEARNING TO DETECT INTERSECTIONAL BIASES

## ABSTRACT

Large Language Models (LLMs) are increasingly deployed in high-stakes domains, yet they often inherit intersectional biases, prejudices that emerge not from a single axis such as race or gender, but from their intersections (e.g., "Black women are too aggressive for leadership"). Existing bias detection and mitigation methods predominantly address single-axis biases and fail to generalize to their complex interactions. In this paper, we present the first unified framework for detecting and mitigating intersectional bias. We construct two paragraph-level intersectional bias dataset: `Indic-Intersect` and `Western-Intersect`, aligned to Indian and Western sociocultural contexts, respectively. For detection, we introduce ***BiasRetriever***, a contrastively trained retriever that learns a bias-aware embedding space by pulling biased text close to canonical stereotypes and pushing it away from unbiased or unrelated examples. BiasRetriever achieves up to $10\%$ more Jaccard score over LLM-based classifiers on unseen intersectional categories and maintains robust cross-domain generalization[1].

## 1 INTRODUCTION

Large language models (LLMs) have rapidly advanced natural language processing and are now widely integrated into decision-making systems, from hiring platforms to healthcare assistants. Yet, these models inherit and amplify the social biases embedded in their training data (Gallegos et al., 2024). Such biases manifest in harmful ways: associating women with domestic roles while linking men to professional careers (Navigli et al., 2023), treating identical résumés differently depending on whether the applicant is Black or White (Gallegos et al., 2024), or producing more toxic completions when prompted with identities linked to marginalized groups (Elsafoury & Katsigiannis, 2024). These disparities highlight a central challenge: *while LLMs appear neutral, their outputs often perpetuate long-standing inequities.*

A particularly pernicious form of bias is *intersectional bias*. Unlike single-axis biases (e.g., only gender or only race), intersectional bias arises at the overlap of multiple identities and produces unique harms (Ma et al., 2023). For instance, stereotypes about Black women cannot be reduced to the sum of racial and gender stereotypes; instead, their experiences reflect distinct and compounded patterns of discrimination. Despite growing recognition in social sciences, computational methods for detecting and mitigating intersectional bias remain scarce. Studies have shown that in AI resume screening, names associated with Black men were selected only $14.8\%$ as often as Black women's names, and $0\%$ compared to White men's names (Wilson & Caliskan, 2024). As LLMs are deployed at an unprecedented scale, these subtle intersectional biases risk reinforcing systemic inequities, making their detection and mitigation a critical to build fair and responsible AI.

Despite a growing body of research in AI fairness, existing paradigms for bias detection and mitigation remain scarce for intersectionality. Many detection methods rely on template-based probes or prompt-based tests, which capture only surface-level stereotypes and struggle to generalize across contexts (Nangia et al., 2020; Souani et al., 2025). Recent work demonstrates that debiasing strategies effective on single dimensions often fail or even amplify harms at intersections (Lalor et al., 2022; Magee et al., 2021). Moreover, multilingual studies show that grammatical gender and cultural context introduce additional challenges (Puttick & Kurpicz-Briki, 2025).

---

[1]We will release the datasets upon acceptance.

Figure 1: **Overview for Intersectional Bias Detection**. We first generate intersectional paragraphs to construct Indic-Intersect and Western-Intersect datasets using SBIC (Sap et al., 2020) and IndiBias (Sahoo et al., 2024) with the help of LLMs. Next, we create a triplet dataset for retriever fine-tuning with triplet loss. After training the retriever, SBIC and IndiBias serve as a database to detect biases in unseen paragraphs.

To address these challenges, we propose a novel framework that detects nuanced intersectional biases in text. We fine-tune a dense retriever using contrastive learning on triplets of (anchor, positive, negative) examples drawn from datasets such as SBIC (Sap et al., 2020) and IndiBias (Sahoo et al., 2024). This trains the *retriever* to learn a **bias-aware embedding space**, where biased paragraphs are mapped close to canonical examples of the same bias and far from unbiased or differently biased text. This enables *sensitive nearest-neighbor detection* of intersectional stereotypes.

**Our contributions are,**

1. We propose a **retriever-based framework for detecting** intersectional bias using contrastive learning over biased/unbiased triplets.
2. We introduce two paragraph-level datasets for intersectional bias detection: **Indic-Intersect** and **Western-Intersect**, comprising a total of 7,404 paragraphs and their corresponding intersectional bias labels.
3. **Empirical Analysis.** Through in-domain and cross-domain validation, showing that **our approach improves bias detection generalization** and produces higher-quality debiased text than strong fine-tuning and prompting baselines.

## 2 RELATED WORK

**Bias in Language Models.** LLMs are known to encode and amplify biases from their training data. Surveys such as Gallegos et al. (2024) provide taxonomies of bias evaluation and mitigation strategies, showing that biases extend beyond statistical imbalance to structural inequities. Navigli et al. (2023) catalog manifestations of bias in pretrained models, with evidence of harms in toxicity detection (Elsafoury & Katsigiannis, 2024) and political discourse modeling (Feng et al., 2023).

**Intersectional Bias.** Compared to single-axis bias, intersectional bias remains underexplored despite its distinct harms. Ma et al. (2023) introduced a dataset for intersectional stereotypes, showing that model behavior diverges from additive effects. Lalor et al. (2022) further demonstrated that debiasing effective on single categories often fails or worsens outcomes on intersectional slices. Complementary studies document intersectional harms in causal LMs (Magee et al., 2021), hate-speech datasets (Kim et al., 2020), and multilingual contexts requiring adapted metrics such as GG-FISE (Puttick & Kurpicz-Briki, 2025). Together, these works highlight the need for explicitly intersectional detection and mitigation methods.

**Bias Detection.** Detection methods include embedding association tests, template-based probes, and prompt-based evaluations (Gallegos et al., 2024; Nangia et al., 2020; Kurita et al., 2019), but these approaches often fail to capture nuanced, context-dependent stereotypes and generalize poorly

across domains. Complementary approaches, such as the clustering-based method LOGAN (Zhao & Chang, 2020), aim to uncover "local" biases within a model's representation space, which often correspond to the unique stereotypes faced by intersectional subgroups. Adaptive prompting offers improvements (Spliethöver et al., 2025), yet suffers from instability. Recent benchmarks emphasize that single-axis probing misses intersectional effects (Lalor et al., 2022; Magee et al., 2021). Recent methods like BiasAlert (Fan et al., 2024) use retrieval in a RAG setup, where LLM is used as a 'judge' to detect biases. In contrast, we train the retriever itself, via contrastive learning as the core detection engine, reframing bias detection as similarity search in a *bias-aware embedding space*. Retriever-based methods trained with contrastive objectives (Kahana & Hoshen, 2022) provide a promising alternative, which we extend for multi-axis bias detection using SBIC (Sap et al., 2020) and IndiBias (Sahoo et al., 2024).

Prior work establishes the pervasiveness of intersectional bias. We advance this literature by combining retriever-based contrastive detection providing the first of its kind framework for detecting intersectional biases.

## 3 INTERSECTIONAL DATASETS FOR BIAS DETECTION

To facilitate a rigorous evaluation of intersectional bias detection, we introduce two novel, large-scale paragraph corpora: **Indic-Intersect**, focusing on sociocultural contexts relevant to India, and **Western-Intersect**, reflecting Western societal contexts. These datasets contain paragraphs centered on specific intersectional identities. A key characteristic of both the corpora is the inclusion of both explicitly biased paragraphs and unbiased paragraphs that share the same intersectional context *but are devoid of prejudice*. The entire data generation process relies on a multi-stage, generative pipeline leveraging Large Language Models (LLMs) for synthesis.

### 3.1 CORPUS CONSTRUCTION

The creation of our final paragraph datasets involved two main stages: (1) the generation of foundational reference corpora of atomic sentences, and (2) the synthesis of complex intersectional paragraphs from these sentences.

#### 3.1.1 REFERENCE CORPORA GENERATION

We first constructed two comprehensive reference corpora of individual sentences, `INDI-REFERENCE` and `SBIC-REFERENCE`, which serve as building blocks for paragraph synthesis.

**Biased Sentence Generation.** For the *Indian context*, we utilized the IndiBias dataset(Sahoo et al., 2024), where bias is often expressed implicitly through comparative sentence pairs. We employed an LLM (*GPT-4o-mini*) to distill each pair into a single, explicitly biased statement, yielding 561 atomic biased sentences. For the *Western context*, we directly sampled 6,972 biased sentences from the SBIC dataset(Sap et al., 2020). The prompt used for this synthesis process is detailed in the Appendix A.8 (see Figure 11).

**Unbiased Sentence Generation.** We created context-aware unbiased sentences. We randomly sampled 140 biased sentences from the IndiBias pool(Sahoo et al., 2024) and 1,700 from the SBIC pool(Sap et al., 2020). An LLM (*meta/llama-4-maverick-17b-128e-instruct*) was prompted to rewrite each sentence to be neutral and inclusive while preserving the original demographic group. The rewritten sentences were labeled as 'unbiased', and the original bias category was stored as 'bias_context'. This process resulted in the final reference corpora: `INDI-REFERENCE` (701 total sentences) and `SBIC-REFERENCE` (8,666 total sentences). A detailed breakdown of the sentence distribution across each bias category is provided in Appendix A.1 (Table 2 and Table 3), and the prompt for this rewriting task is shown in Figure 12.

#### 3.1.2 INTERSECTIONAL PARAGRAPH SYNTHESIS

The final paragraph corpora were generated by synthesizing narratives from the reference sentences. The intersectional categories, comprising combinations of two or three distinct bias

axes, were predefined by the authors to ensure narrative coherence. This resulted in 3700 paragraphs from 18 categories for `Indic-Intersect` and 3704 paragraphs from 20 categories for `Western-Intersect`. The final distribution across these categories (Table 4) and dataset examples of paragraphs (Table 5 and Table 6) for both corpora is detailed in Appendix A.2.

**Biased Paragraphs.** For each intersectional category (e.g., "gender+caste"), we sampled one biased sentence per constituent axis from the reference corpora. These sentences served a dual purpose in a prompt for an LLM (*GPT-4o-mini*): key identity attributes were extracted to form a character profile, and the sentences acted as contextual seeds. The LLM was tasked with weaving these into a cohesive narrative depicting an experience of prejudice, with a "show, not tell" constraint, as detailed in the Appendix (Figure 13 and 14).

**Unbiased Paragraphs with Coherence-Based Sampling.** To generate challenging unbiased examples, we developed a semantic coherence-based sampling strategy. For a given intersectional context, we sampled a candidate pool of unbiased sentences from the reference corpora. These were encoded into embeddings using a Sentence-BERT model (*all-MiniLM-L6-v2*). We then performed an optimization search to find a combination of sentences (one per axis) that maximized the average pair-wise cosine similarity, ensuring thematic alignment. This coherent set of sentences was then provided to an LLM (*meta/llama-4-maverick-17b-128e-instruct*) with instructions to generate a neutral-to-positive, slice-of-life narrative explicitly avoiding any depiction of prejudice. The specific prompts used for the `Indic-Intersect` and `Western-Intersect` datasets are detailed in the Appendix (see Figure 15 and Figure 16).

### 3.2 HUMAN VALIDATION AND ANNOTATION

Given the multi-stage, LLM-driven nature of our data generation, we implemented a comprehensive human validation process to ensure the quality and fidelity of the assets at each stage. This process was divided into two phases: validation of the foundational reference sentences and validation of the final intersectional paragraphs. All annotation tasks were performed by **two trained annotators** familiar with sociocultural nuances relevant to each corpus.

#### 3.2.1 VALIDATION OF REFERENCE CORPORA

First, we validated the quality of the atomic sentences in the `INDI-REFERENCE` and `SBIC-REFERENCE` corpora. A random sample of 800 sentences from `SBIC-REFERENCE` and the complete `INDI-REFERENCE` were used for evaluation. The annotators' primary task was to verify the correctness of the sentence labels. For sentences labeled "biased", they confirmed the presence of explicit prejudice. For sentences rewritten to be "unbiased", they verified that the sentence was neutral or positive while successfully preserving the original demographic context. We achieved a **Cohen's Kappa of** $\kappa = 0.91$ for this sentence-level labeling task, indicating almost perfect agreement. This high level of agreement affirmed the reliability of our sentence generation and rewriting pipeline, ensuring we had high-quality building blocks for paragraph synthesis. In cases of annotator disagreement, we conducted adjudication through joint discussion to create the final reference sentence.

#### 3.2.2 VALIDATION OF INTERSECTIONAL PARAGRAPHS

Next, we validated the final `Indic-Intersect` and `Western-Intersect` paragraph corpora. For this phase, we randomly sampled 400 paragraphs from each dataset. The annotators performed a multi-faceted evaluation for each paragraph:

1. **Bias Label Verification:** Confirming the overall narrative was correctly labeled as "biased" (depicting prejudice) or "unbiased" (a neutral/positive slice-of-life story).
2. **Intersectional Identity Check:** Verifying that the intended intersectional identities (e.g., "gender+caste") were clearly and coherently represented within the narrative.
3. **Quality Assessment:** Rating the paragraph's grammatical correctness, fluency, and narrative plausibility on a 3-point Likert scale.

For the primary task of bias label verification at the paragraph level, we observed a **Cohen's Kappa of** $\kappa = 0.89$. The slightly lower, yet still substantial, agreement compared to the sentence-level task

reflects the increased complexity of interpreting bias in longer, narrative contexts. All disagreements in both validation phases were resolved by the authors to establish a final ground truth. These rigorous validation results confirm the high quality and reliability of our final datasets for the study of intersectional bias.

### 3.3 DATA PARTITIONING

We split both the corpora into two two different sets to inspect the model's ability to generalize to unseen intersectional combinations. We defined three distinct, predefined subsets of training categories, denoted as $C_1, C_2, C_3$. Let the full dataset be $D = D_{biased} \cup D_{unbiased}$. The experiment corresponding to each subset $C_i$ has:

1. a biased set, $D_{biased}$, which is partitioned into $D_{C_i}$ (paragraphs in seen categories) and $D_{N-C_i}$ (paragraphs in unseen categories), where $N$ is the total number of intersectional categories in the corpora. $\{N = 18$ for `Indic-Intersect`, $N = 20$ for `Western-Intersect`$\}$

2. a training set, $D_{train,C_i}$, which is constructed from a stratified $80\%$ split of $D_{C_i}$ and a $70\%$ split of $D_{unbiased}$.

We form three different test sets. All the three sets include $30\%$ paragraphs of $D_{unbiased}$ which are not used during training and the following:

1. $D_{test,C_i}$ (for seen categories): Stratified $20\%$ paragraphs from $C_i$ which are different from the corresponding $D_{train,C_i}$.

2. $D_{test,N-C_i}$ (for unseen categories): All paragraphs from $N - C_i$ categories. These paragraphs are not used during training using $C_i$.

3. $D_{test,N}$ (a comprehensive set of all held-out data): The combination of $D_{test,C_i}$ and $D_{test,N-C_i}$.

This ensures no data leakage between training and evaluation. The specific composition of each subset $C_i$ for both corpora is detailed in the Appendix (see Table 7).

### 3.4 TRIPLET MINING FOR RETRIEVER FINE-TUNING

To adapt the semantic space of our retriever to the specific nuances of intersectional bias, we fine-tune it using a Triplet Loss objective. This requires a large-scale, high-quality dataset of triplets, $\mathcal{T} = (\mathcal{A}, \mathcal{P}, \mathcal{N})$, where each triplet consists of an anchor $(a)$, a positive sample $(s_\mathcal{P})$ semantically related to the anchor, and a negative sample $(s_\mathcal{N})$ that is unrelated. We developed several distinct settings for triplet curation to analyze the impact of data quantity, quality, and generation strategy on model performance.

**Anchor and Sample Definitions.** The anchor $(a)$ in our triplet construction is always a full paragraph from either the `Indic-Intersect` or `Western-Intersect` corpus. Anchors can be either biased $(a_\mathcal{B})$ or unbiased $(a_\mathcal{U})$. For a given anchor, the positive $(s_\mathcal{P})$ and negative $(s_\mathcal{N})$ samples are individual sentences either sourced from our `INDI-REFERENCE` and `SBIC-REFERENCE` corpora using the strategies outlined below or generated using LLM (*meta/llama-4-maverick-17b-128e-instruct*).

**Selection Criteria.** Our primary method for sample selection is Semantic Retrieval. Using a FAISS-indexed vector library of our reference corpora, we mine for "hard" positive and negative samples.

- **For a biased anchor** $(a_\mathcal{B})$ with constituent bias labels $L$, a positive sample $(s_\mathcal{P})$ is a sentence retrieved from the reference corpus that shares a label $l_i \in L$ and is semantically close to $a_\mathcal{B}$. A negative sample $(s_\mathcal{N})$ is a sentence that is also semantically close to $\mathcal{A}_b$ but has an unrelated label $(l_j \notin L)$.
- **For an unbiased anchor** $(a_\mathcal{U})$ with intersectional context $C_p$, a positive sample $(s_\mathcal{P})$ is a semantically similar sentence also labeled 'unbiased'. A negative sample $(s_\mathcal{N})$ is a biased sentence whose bias label is one of the anchor's contextual categories $(l_j \in C_p)$.

## 3.5 TRIPLET CURATION SETTINGS

We construct four distinct master triplet corpora based on variations in retrieval depth ($k$) and the composition of the negative pool.

- **SR-k4 (Semantic Retrieval, k=4):** Our baseline triplet set. For each anchor, we retrieve the top $k = 4$ positives and pair them rank-wise with the top $k = 4$ negatives. For biased anchors, the negative pool consists exclusively of sentences with other bias labels which does not belong to constituent bias labels of anchor ($L_p$).

- **SR-k10 (Semantic Retrieval, k=10):** This setting is identical to SR-k4 but increases the retrieval depth to $k = 10$, creating a larger training set to test the impact of data quantity and retriever training efficiency.

- **SR-k4-UN (SR with Unbiased Negatives):** This setting modifies the negative sampling strategy for biased anchors ($p_B$). The negative pool is composed of sentences from other biased categories (80% probability) and the 'unbiased' category (20% probability). This creates more robust, yet challenging triplets, forcing the model to differentiate biased content from neutral discussion of the same topics.

- **SR-k4+LLM (SR Augmented with LLM):** We begin with the SR-k4 triplet set and augment it by concatenating a large corpus of ∼20,000 triplets generated via our three LLM-based strategies A.4 (Dual-LLM Generation, LLM-Positive with Counterfactual Negative, Mined-Positive with Counterfactual Negative, Neutral Anchor Paraphrasing). This setting evaluates the benefit of adding synthetically generated, diverse training examples.

Each of these four master triplet corpora is generated independently for both `Indic-Intersect` and `Western-Intersect`. The resulting triplet quantities for each setting are summarized in Table 1.

Table 1: Total number of triplets generated for each curation

| Curation | Indic-Intersect | Western-Intersect |
|---|---|---|
| SR-k4 | 31,568 | 31,140 |
| SR-k10 | 78,920 | 105,070 |
| SR-k4-UN | 31,568 | 31,140 |
| SR-k4+LLM | 51,568 | 51,140 |

To ensure a direct and equitable comparison with our BERT-based baseline, we fine-tune and evaluate the retriever under an identical experimental design as described in Section 3.3. Each of the four master triplet dataset described above is filtered to create dedicated training sets for the $C_1, C_2$, and $C_3$ experimental runs. This filtering ensures that for any given run $C_i$, the retriever is only fine-tuned on triplets whose anchors belong to the categories within $C_i$ and do not appear in the test set $D_{test,C_i}$. A retriever fine-tuned on a specific triplet dataset $D_{train,C_i}$ is subsequently evaluated on the test sets ($D_{test,C_i}$, $D_{test,N-C_i}$, or $D_{test,N}$) of same corpora or different corpora, allowing for a rigorous comparison of generalization capabilities.

## 4 INTERSECTIONAL BIAS DETECTION

Our proposed methodology for intersectional bias detection is centered around a retrieval-based framework. We hypothesize that an intersectionally biased text can be effectively identified by its semantic proximity to multiple texts, each exhibiting one of the constituent biases. This section details our approach, including the baseline models for comparison, the architecture and training of our proposed *Bias-Aware* Retriever, and the specific datasets used in our experiments.

### 4.1 PROBLEM STATEMENT

Let $\mathcal{X}$ denote the space of textual inputs, where each paragraph $x \in \mathcal{X}$ may contain one or more categories of social bias. Our objective is to learn a retrieval model $\mathcal{R}_\theta : \mathcal{X} \to \mathbb{R}^d$, parameterized by $\theta$, that maps paragraphs into a bias-aware embedding space. In this space, intersectionally biased inputs are pulled close to canonical examples of their constituent single-bias categories and pushed far from unrelated or unbiased samples.

**Algorithm 1** Intersectional Bias Retriever

---

**Require:** Triplet dataset $\mathcal{T}$, pretrained encoder $\mathcal{R}_\theta$, batch size $B$, epochs $E$
1: **for** $e = 1$ to $E$ **do**
2:     Sample batch $\{(a_i, s_{\mathcal{P}i}, s_{\mathcal{N}i})\}_{i=1}^{B}$ from $\mathcal{T}$
3:     Encode: $h_a \leftarrow \mathcal{R}_\theta(a_i)$, $h_p \leftarrow \mathcal{R}_\theta(s_{\mathcal{P}i})$, $h_n \leftarrow \mathcal{R}_\theta(s_{\mathcal{N}i})$
4:     Compute similarity scores: $s_{ap} \leftarrow \mathrm{sim}(h_a, h_p)$, $s_{an} \leftarrow \mathrm{sim}(h_a, h_n)$
5:     Compute loss $\mathcal{L}$ via Multiple Negatives Ranking Loss
6:     Update $\theta \leftarrow \theta - \eta \nabla_\theta \mathcal{L}$
7: **end for**
8: **return** Trained retriever $\mathcal{R}_\theta$

---

**Hypothesis Discussion.** Our hypothesis is that a contrastively trained retriever can exploit the compositional structure of intersectional bias, outperforming classification-based approaches in two settings: (i) **in-domain generalization**, detecting previously unseen bias categories within the same source dataset, and (ii) **cross-domain transfer**, detecting intersectional bias when applied to out-of-distribution datasets. Intuitively, a paragraph expressing sexism + racism should embed near single-bias exemplars of both dimensions. In practice, this means that a query paragraph embedding will retrieve sentences carrying the same bias signatures, even if their exact intersection has never been observed during training.

For example, consider the sentence: *"Black women are too aggressive for leadership roles."* This statement reflects both racial and gender bias. A well-trained retriever should place it near sentences containing racial stereotypes (e.g., *"Black people are unfit for professional settings"*) and gender stereotypes (e.g., *"Women should not lead companies"*), while keeping it distant from unrelated biases (e.g., stereotypes about religion) or unbiased text. Unlike conventional classifiers that assign discrete labels, our retrieval-based formulation supports fine-grained similarity search and enables generalization to unseen bias categories and out-of-distribution datasets.

### 4.2 BIASRETRIEVER: A *Bias-Aware* RETRIEVER TRAINING

Given a dataset $\mathcal{T} = (\mathcal{A}, \mathcal{P}, \mathcal{N})$, we fine-tune a transformer-based retriever $\mathcal{R}_\theta$ (initialized from `msmarco-roberta-base-v2`[2]) using Multiple Negatives Ranking Loss (Henderson et al., 2017). For a triplet $(a, s_\mathcal{P}, s_\mathcal{N})$, we obtain embeddings $h_a = \mathcal{R}_\theta(a)$, $h_p = \mathcal{R}_\theta(s_\mathcal{P})$, and $h_n = \mathcal{R}_\theta(s_\mathcal{N})$. The loss is defined as:

$$\mathcal{L} = -\log \frac{\exp(\mathrm{sim}(h_a, h_{s_\mathcal{P}})/\tau)}{\exp(\mathrm{sim}(h_a, h_{s_\mathcal{P}})/\tau) + \exp(\mathrm{sim}(h_a, h_{s_\mathcal{N}})/\tau)},$$

where $\mathrm{sim}(\cdot, \cdot)$ is cosine similarity, $\tau$ is a temperature parameter, and $\mathcal{N}$ is the set of negatives in the batch. This objective encourages anchors to cluster with bias-matched positives while repelling unrelated negatives.

The end-to-end training process is summarized in Algorithm 1, which formalizes our retriever optimization loop. We train four different models for both the corpora, each corresponding to four different curate datasets described in 3.5. We call these models BiasRetriever (`B4_UB4`), BiasRetriever (`B10_UB10`), BiasRetriever (`B4_NUB_UB4`), BiasRetriever (LLM), respectively for the four settings.

### 4.3 BASELINES

We compare our retriever-based detection against two baselines: 1. **Classifier Baseline** (BERT-MultiLabel): Fine-tuning BERT on $\mathcal{D}_{train_{C_i}}$, a paragraph-level multi-label classification dataset with cross-entropy loss, to predict the presence of multiple bias categories. 2. **Unsupervised Retriever Baseline** (S-BERT-Base): Evaluating the frozen base retriever (`msmarco-roberta-base-v2`) without fine-tuning on our triplet data.

---

[2]https://huggingface.co/sentence-transformers/msmarco-roberta-base-v2

## 5 RESULTS AND ANALYSIS

To evaluate the effectiveness of our proposed retrieval-based approach for intersectional bias detection, we conduct a series of experiments comparing our fine-tuned **BiasRetriever** against strong baselines across multiple challenging scenarios.

### 5.1 EXPERIMENTAL SETUP

**Evaluation Scenarios.**   We assess model performance in four distinct settings to rigorously test for both in-domain and out-of-domain generalization:

- **In-Domain:** (1) Train on *Indic-Intersect* triplets, test on *Indic-Intersect*. (2) Train on *Western-Intersect* triplets, test on *Western-Intersect*.
- **Out-of-Domain (OOD):** (1) Train on *Indic-Intersect*, test on *Western-Intersect*. (2) Train on *Western-Intersect*, test on *Indic-Intersect*.

**Retrieval Process.**   For each paragraph in the test set, our retriever-based models query a reference corpus to fetch the top-$k$ most semantically similar sentences. The database for *Indic-Intersect* is the *INDI-REFERENCE* corpus, and for *Western-Intersect*, it is the *SBIC-REFERENCE* corpus. The predicted bias categories for the paragraph are determined by aggregating the bias labels of these top-$k$ retrieved sentences.

**Evaluation Metric.**   We primarily use the **Jaccard Similarity** to measure the overlap between the set of predicted bias categories ($C_{pred}$) and the set of ground-truth bias categories ($C_{true}$). This metric is well-suited for multi-label tasks as it penalizes both false positives and false negatives. It is defined as:

$$J(C_{true}, C_{pred}) = \frac{|C_{true} \cap C_{pred}|}{|C_{true} \cup C_{pred}|}$$

We also report **Exact Match**, a stricter metric that is 1 if $C_{pred} = C_{true}$ and 0 otherwise.

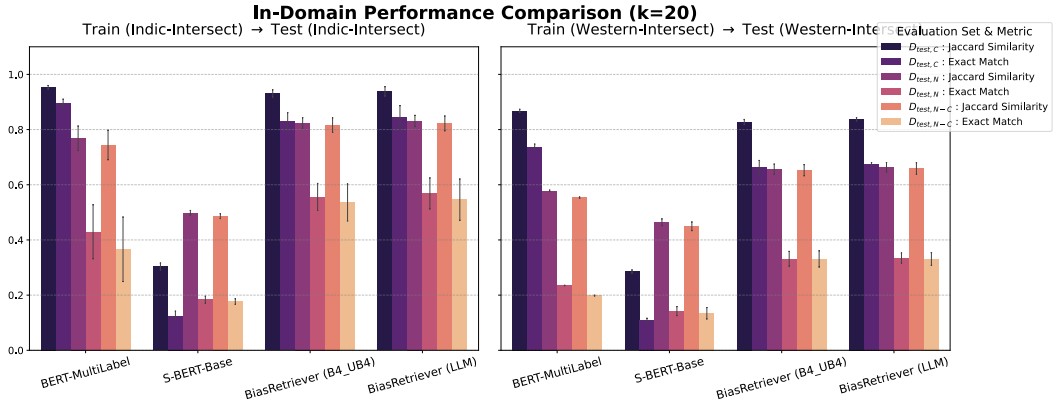

Figure 2: Performance comparison across **In-Domain** scenarios. Our `BiasRetriever` models consistently outperform baselines, especially on the unseen intersectional categories ($D_{test,N-C_i}$) for both corpora, demonstrating superior generalization. The error bars represent standard deviation across three subsets, $C_1, C_2, C_3$.

### 5.2 MAIN RESULTS: GENERALIZATION PERFORMANCE

Figures 2 and 3 provide a comprehensive comparison of model performance. The results highlight our method's significant advantages, particularly in generalizing to unseen intersectional identities.

In the **in-domain setting** (Figure 2), we observe that while the `BERT-MultiLabel` baseline performs well on intersectional categories seen during training ($D_{test,C_i}$), its performance collapses

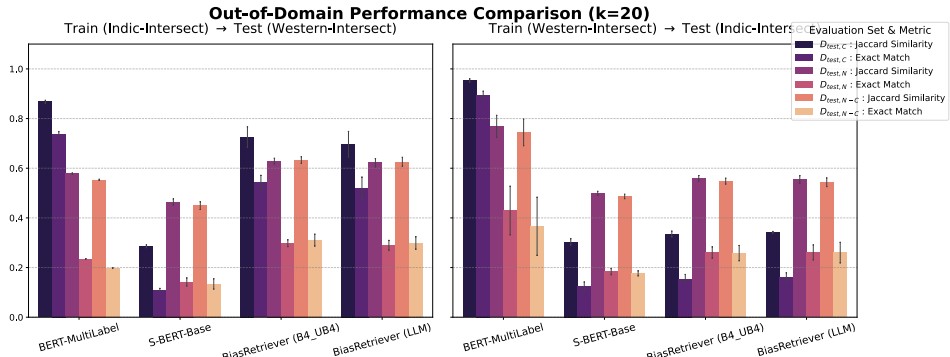

Figure 3: **Cross-domain transfer performance**. Models trained on Indic-Intersect tested on Western-Intersect and vice versa. `BiasRetriever` maintains robustness across domains, unlike classification baselines that degrade sharply.

when faced with unseen intersections ($D_{test,N-C_i}$). This demonstrates a classic failure to generalize. The `S-BERT-Base` retriever is more stable but achieves low scores across the board. In contrast, our `BiasRetriever` models, particularly the LLM-trained variant, exhibit high and remarkably stable performance across both seen and unseen categories. This indicates that our contrastive fine-tuning enables the model to learn a compositional understanding of bias that generalizes effectively to novel combinations within the same cultural context.

The out-of-domain setting (Figure 3) presents the most challenging test of robustness. Importantly, the `BERT-MultiLabel` baseline cannot be meaningfully applied in a cross-domain setup since the label space differs between datasets, making direct transfer infeasible. By contrast, our retrieval-based formulation is label-agnostic, enabling natural transfer. We further observe an asymmetry in cross-domain gains: models trained on `Indic-Intersect` transfer more effectively to `Western-Intersect` than the reverse. A likely explanation is that `Western-Intersect` is grounded in SBIC, whose instances are short, noisy social media posts; these do not serve as strong retrieval database when used for the more coherent, paragraph-level narratives of `Indic-Intersect`. Finally, a striking result is that for `Western-Intersect`, cross-domain training on `Indic-Intersect` yields even higher performance than the `BERT-MultiLabel` model achieves when trained and evaluated on `Western-Intersect` itself. This underscores the robustness of the `BiasRetriever` embeddings and their ability to capture domain-agnostic structures of bias.

Our analysis of retrieval depth shows that while increasing k generally boosts performance, gains plateau beyond k = 15 in the in-domain setting, and performance even declines at higher depths on Western-Intersect due to noisy low-quality retrievals. Importantly, BiasRetriever consistently outperforms frozen S-BERT-Base across all k, highlighting that improvements stem from contrastive fine-tuning rather than simply retrieving more documents. An ablation study further reveals a hierarchy in triplet generation strategies: the baseline B4_UB4 is solid, adding hard negatives improves results, and the LLM Augmented strategy achieves the best performance, underscoring the value of diverse, high-quality synthetic triplets for learning robust and generalizable representations of intersectional bias. We discuss these results in depth in Appendix A.5 and A.6.

## 6 CONCLUSION AND FUTURE WORK

In this work, we introduced `BiasRetriever`, a contrastively trained retriever for intersectional bias detection. Through experiments on the newly constructed `Indic-Intersect` and `Western-Intersect` corpora, we showed that retrieval-based detection generalizes significantly better than classification baselines, both to unseen bias combinations and to out-of-domain settings. These results highlight the strength of retrieval as a label-agnostic formulation for modeling compositional structures of bias.

AUTHOR CONTRIBUTIONS

If you'd like to, you may include a section for author contributions as is done in many journals. This is optional and at the discretion of the authors.

ACKNOWLEDGMENTS

Use unnumbered third level headings for the acknowledgments. All acknowledgments, including those to funding agencies, go at the end of the paper.

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

# A    CLASSIFIER WITH DIFFERENTIABLE HAMMING LOSS

As an alternative to the conventional cross-entropy loss, we introduce a second classifier baseline trained with a differentiable approximation of the **Hamming loss**. While binary cross-entropy independently optimizes the log-likelihood for each label, the Hamming loss offers a different objective that is often more aligned with the goals of multi-label classification. It directly measures the fraction of all labels that are misclassified, treating false positives and false negatives with equal importance. In the context of intersectional bias, this is particularly relevant as the goal is to identify the precise **set of intersectional biases**, making the total number of label errors a more intuitive performance measure.

The standard Hamming loss for a single sample with $L$ labels is defined as the fraction of incorrect predictions. Given a ground-truth binary vector $\mathbf{y} \in \{0,1\}^L$ and a predicted binary vector $\hat{\mathbf{y}} \in \{0,1\}^L$, the loss is given by $\frac{1}{L} \sum_{i=1}^{L} \mathbf{y}_i \oplus \hat{\mathbf{y}}_i$, where $\oplus$ denotes the XOR operation. However, this formulation is non-differentiable due to the hard thresholding required to obtain $\hat{\mathbf{y}}$ from the model's raw logits, $\mathbf{z}$. To address this, we implement a differentiable surrogate loss. We first compute the class probabilities using a temperature-scaled sigmoid function, $\mathbf{p} = \sigma(\mathbf{z}/\tau)$, and then define the loss as:

$$\mathcal{L}_H(\mathbf{z}, \mathbf{y}) = \frac{1}{L} \sum_{i=1}^{L} \left[ p_i(1 - y_i) + (1 - p_i)y_i \right] \tag{1}$$

This function serves as a smooth proxy for the discrete Hamming loss by directly penalizing the probability assigned to the incorrect class for each label. The temperature parameter, $\tau$, controls the sharpness of the sigmoid output, which can help in learning better-calibrated probabilities.

For this baseline, we fine-tune a `ModernBERT-base` model on the same multi-label dataset $\mathcal{D}_{train,C_i}$ used for the cross-entropy classifier to ensure a fair comparison. The model is trained for a maximum of 20 epochs using the **AdamW optimizer** with a learning rate of $2 \times 10^{-5}$ and early stopping with a patience of 3. We use the differentiable Hamming loss with a temperature of $\tau = 0.7$. This experimental setup allows us to directly evaluate the impact of the loss function on the classifier's ability to generalize to both seen and unseen intersectional categories.

## A.1    REFERENCE DATASET DETAILS AND CATEGORY DISTRIBUTIONS

Here we provide a detailed breakdown of the sentence distribution within our foundational reference corpora.

Table 2: Sentence distribution in the `INDI-REFERENCE` corpus.

| Label | Count |
|---|---|
| gender | 197 |
| unbiased | 140 |
| socioeconomic | 108 |
| religion | 81 |
| age | 62 |
| caste | 48 |
| physical-appearance | 41 |
| disability | 24 |

Table 3: Sentence distribution in the `SBIC-REFERENCE` corpus.

| Label | Count |
|---|---|
| unbiased | 1694 |
| race | 1540 |
| gender | 1540 |
| social | 901 |
| culture | 800 |
| victim | 800 |
| disabled | 800 |
| body | 591 |

## A.2    PARAGRAPH DATASET DISTRIBUTIONS AND DATASET EXAMPLES

Here we provide a detailed breakdown of the paragraph distribution within our final, synthesized corpora.

Table 4: Paragraph counts per category in the `Indic-Intersect` (Total: 3,700) and `Western-Intersect` (Total: 3,704) datasets.

Indic-Intersect

| Label | Count |
|---|---|
| unbiased | 900 |
| gender+socioeconomic | 298 |
| G+S+R | 224 |
| gender+religion | 224 |
| religion+socioeconomic | 224 |
| G+A+S | 171 |
| gender+age | 171 |
| G+A+R | 171 |
| G+C+S | 133 |
| caste+socioeconomic | 133 |
| G+C+A | 133 |
| gender+caste | 133 |
| gender+P | 113 |
| G+R+P | 112 |
| G+C+P | 112 |
| C+S+P | 112 |
| caste+P-appearance | 112 |
| G+A+P | 112 |
| G+S+P | 112 |
| **Total** | **3,700** |

Western-Intersect

| Label | Count |
|---|---|
| unbiased | 982 |
| CUL+G+SOC | 140 |
| G+DIS+SOC | 140 |
| gender+social | 140 |
| disabled+social | 139 |
| B+G+CUL | 139 |
| culture+gender | 139 |
| race+social | 138 |
| CUL+R+SOC | 138 |
| body+race | 137 |
| gender+disabled | 136 |
| culture+race | 136 |
| body+gender | 135 |
| G+R+SOC | 135 |
| G+B+DIS | 134 |
| victim+race | 134 |
| R+B+SOC | 134 |
| V+SOC+R | 134 |
| DIS+R+CUL | 133 |
| V+G+R | 132 |
| gender+race | 129 |
| **Total** | **3,704** |

*Note: For brevity in the tables, categories are abbreviated as follows: G (gender), R (religion), C (caste), S (socioeconomic), A (age), P (physical-appearance), CUL (culture), SOC (social), DIS (disabled), B (body), and V (victim).*

Here we provide illustrative examples from our two main corpora, showcasing a matched pair of biased and unbiased paragraph.

Table 5: An example from the `Indic-Intersect` dataset.

| Paragraph | Label |
| --- | --- |
| *Amina walked into the crowded café, her hijab catching the eye of a few patrons. As she waited in line, she overheard a man whisper, "Women often panic in dangerous situations, while men manage to stay composed." She clenched her jaw, knowing that people often underestimated her strength. When she finally reached the counter, the barista hesitated, glancing at her muscular arms, as if unsure whether to serve her. "Are you sure you can handle the heat of this coffee?" he joked, smirking. Amina felt the sting of his words, realizing some still believed Muslims struggle to adapt to modern society due to their outdated beliefs.* | `gender+age` `+physical-appearance` |
| *As a young Adivasi woman, Rohini beamed with pride as she watched her overweight teenage nephew, Kaito, effortlessly push a heavy cart filled with fresh produce at their family's weekly market stall. Kaito's strength was a valuable asset, and he took great joy in helping their community by providing for the local households. Nearby, Rohini's elderly neighbor, a wise Brahmin woman named Leela, chatted with customers while expertly arranging vibrant flowers. The warmth and camaraderie among them was palpable, filling the market with a sense of community and belonging.* | `unbiased` |

Table 6: An example from the `Western-Intersect` dataset.

| Paragraph | Label |
| --- | --- |
| *[In a bustling café in central Cairo, Layla struggled to order her favorite coffee while the barista ignored her in favor of a group of men discussing video games. "Until the Middle East has their own version of Pac-Man, it's called Bomberman," one laughed, dismissing the women at the table. Layla sighed, her heart heavy with the weight of familiar frustration as she overheard another patron joke, "What's similar between women's rights and weapons of mass destruction? You won't find either in the Middle East." The laughter stung, a reminder of her daily battle against a world that often rendered her invisible, her voice unheard.* | `gender+culture` |
| *At the community center's annual block party, DeShawn, a talented young rapper, took the stage, impressing the crowd with his lyrics. Nearby, Allen, a frat brother, was volunteering at a charity booth, discussing philanthropy with locals. As they worked together, they discovered a shared passion for Japanese culture and music. Lil' Boy, a local DJ, spun a set that got everyone dancing, including a group of friends from different walks of life, all enjoying the vibrant atmosphere. The community came together, celebrating their diversity and the talents that made their neighborhood thrive.* | `unbiased` |

## A.3 COMPOSITION OF C SUBSETS FOR GENERALIZATION EXPERIMENTS

Table 7: Predefined subsets of training categories used for the C-Category (Zero-Shot Generalization) experiments. Each model is trained on one subset ($C_i$) and evaluated on the remaining ($N - C_i$) categories.

| Corpus | Subset | Intersectional Categories |
|--------|--------|---------------------------|
| Indic-Intersect | $C_1$ | age+caste+gender, age+gender, age+gender+physical-appearance, age+gender+socioeconomic, caste+gender+socioeconomic, caste+physical-appearance, gender+physical-appearance+socioeconomic, gender+religion, religion+socioeconomic |
| | $C_2$ | caste+gender, caste+gender+physical-appearance, caste+physical-appearance, caste+socioeconomic, age+gender+religion, age+gender+socioeconomic, gender+religion, gender+physical-appearance+religion, gender+religion+socioeconomic |
| | $C_3$ | caste+socioeconomic, caste+physical-appearance+socioeconomic, age+gender, age+gender+religion, age+caste+gender, caste+gender+physical-appearance, gender+physical-appearance, gender+socioeconomic, religion+socioeconomic |
| Western-Intersect | $C_1$ | body+gender, culture+disabled+race, culture+gender, culture+gender+social, gender+disabled, disabled+social, gender+race, gender+race+social, victim+gender+race, race+social |
| | $C_2$ | body+race, race+body+social, culture+gender, culture+race, culture+race+social, gender+disabled+social, gender+race+social, gender+social, race+social, victim+social+race |
| | $C_3$ | body+gender+culture, gender+body+disabled, body+gender, culture+race, gender+disabled, disabled+social, gender+disabled+social, gender+social, victim+social+race, victim+race |

## A.4 LLM-BASED TRIPLET AUGMENTATION STRATEGIES

To augment our training data, we employ an LLM, $\mathcal{G}(\cdot)$, to generate novel triplets using four distinct strategies. These strategies are summarized in Table below.

Table 8: Summary of LLM-based triplet generation strategies.

| Strategy | Anchor ($A$) | Positive ($P$) Method | Negative ($N$) Method | Purpose |
|---|---|---|---|---|
| **Dual-LLM Generation** | Biased Paragraph ($p_B$) | Generated by LLM: $s_{P_{LLM}} = \mathcal{G}(\text{prompt}_{pos}17, p_B)$ | Generated by LLM: $s_{N_{LLM}} = \mathcal{G}(\text{prompt}_{neg}18, p_B)$ | To generate completely novel triplets where both positive and negative are thematically linked to the anchor. |
| **LLM-Positive with Counterfactual Negative** | Biased Paragraph ($p_B$) | Generated by LLM: $s_{P_{LLM}} = \mathcal{G}(\text{prompt}_{pos}17, p_B)$ | Generated by LLM via counterfactual transformation of the positive: $s_{N_{CF}} = \mathcal{G}(\text{prompt}_{cf}19, s_{P_{SR}})$ | To pair novel, LLM-generated positives with challenging, semantically-close negatives from the existing corpus. |
| **Mined-Positive with Counterfactual Negative** | Biased Paragraph ($p_B$) | Mined via SR: $s_{P_{SR}}$ | Generated by LLM via counterfactual transformation of the positive: $s_{N_{CF}}19 = \mathcal{G}(\text{prompt}_{cf}, s_{P_{SR}})$ | To create maximally difficult negatives that are structurally identical to positives, forcing the model to learn fine-grained distinctions. |
| **Neutral Anchor Paraphrasing** | Unbiased Paragraph ($p_U$) | Generated by LLM by paraphrasing the anchor: $s_{P_{LLM}} = \mathcal{G}(\text{prompt}_{para}20, p_U)$ | Sampled via SR from a pool of unbiased sentences with a disjoint context. | To teach the model to associate neutral paragraphs with other neutral phrasings while distinguishing them from unrelated neutral topics. |

## A.5 ANALYSIS OF RETRIEVAL PARAMETER K

To assess the effect of retrieval depth, we analyze model performance on the overall test set ($D_{test,N}$) as a function of $k$ (Figures 4 and 5). While increasing $k$ generally improves performance across retriever models, two important trends emerge. First, in the in-domain setting, the gains diminish beyond $k = 15$, suggesting that the embedding space saturates once sufficient contextual evidence has been retrieved. Second, on Western-Intersect, increasing from B4_UB4 to B10_UB10 actually causes a drop in performance, likely due to noise from low-quality social media sentences retrieved at higher depths. Most crucially, across all values of $k$, there remains a substantial and consistent gap between our BiasRetriever variants and the frozen S-BERT-Base, underscoring that the improvements stem from our contrastive fine-tuning rather than from simply retrieving more documents.

## A.6 ABLATION STUDY: TRIPLET GENERATION STRATEGIES

Finally, we conduct an ablation study to evaluate the effectiveness of our different triplet generation strategies. Figure 6 compares the in-domain performance of models trained using each strategy at $k = 20$.

The results, which are remarkably consistent across both training domains, reveal a clear performance hierarchy. The B4_UB4 strategy serves as a solid baseline. Introducing hard negatives via the B4_NUB_UB4 strategy provides a noticeable improvement. Critically, the LLM Augmented strategy emerges as the clear winner, achieving the highest scores across all conditions. This provides strong evidence that training on diverse, high-quality, synthetically generated triplets is the most effective approach for learning a robust and highly generalizable representation of intersectional bias.

## A.7 ABLATION EXPERIMENTS WITH ANOTHER RETRIVER: MINILM

Through figure 7 and 8 we show the effectiveness of our algorithmic using another retriever, this shows our retriever is not model agnostic.

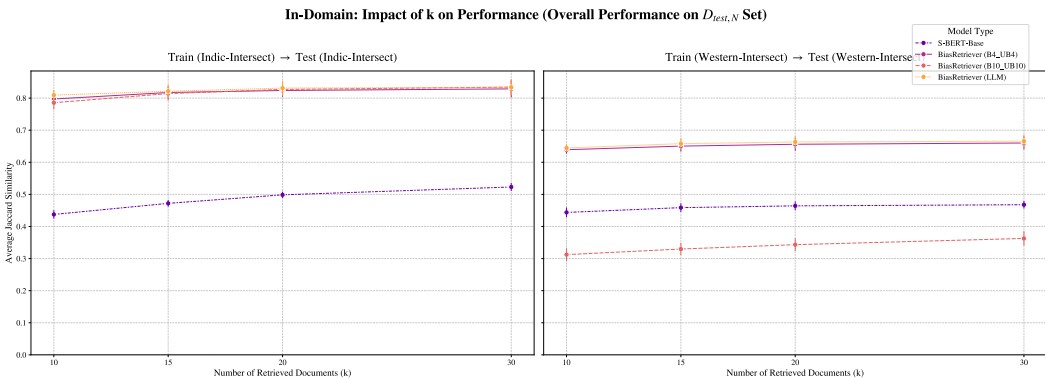

Figure 4: **In-domain impact of retrieval depth.** Larger $k$ improves recall, showing the benefit of more positives and negatives.

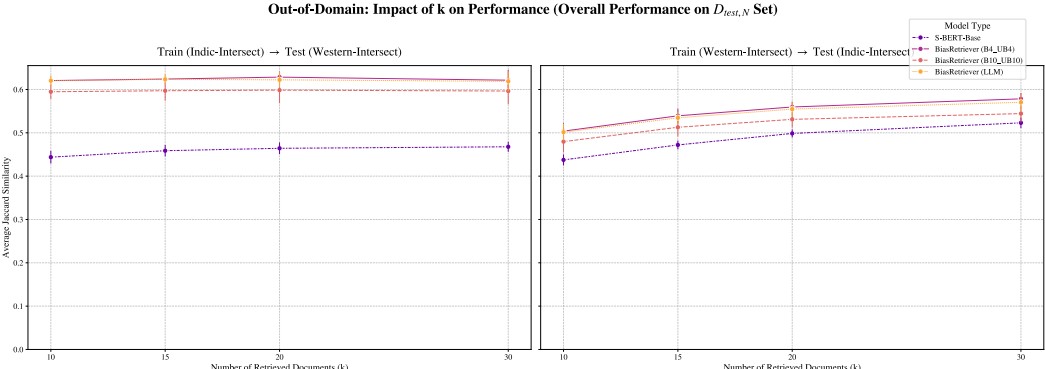

Figure 5: **Cross-domain impact of retrieval depth**.

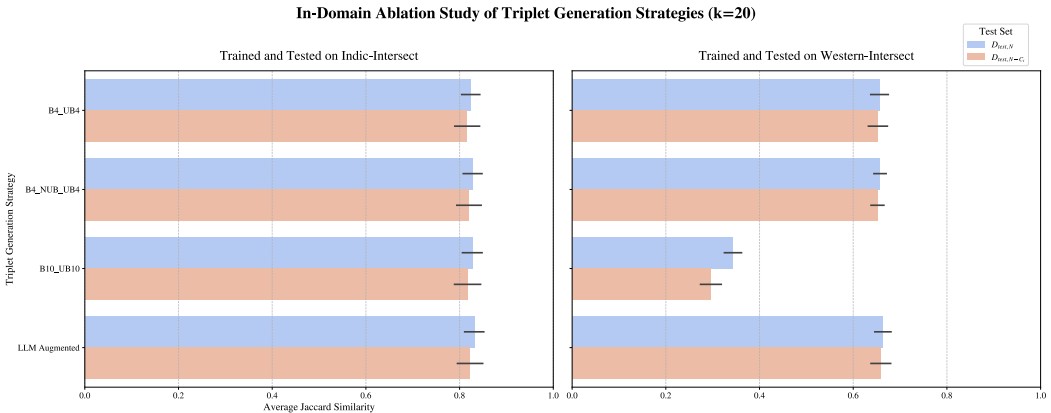

Figure 6: Comparison of Triplet strategy

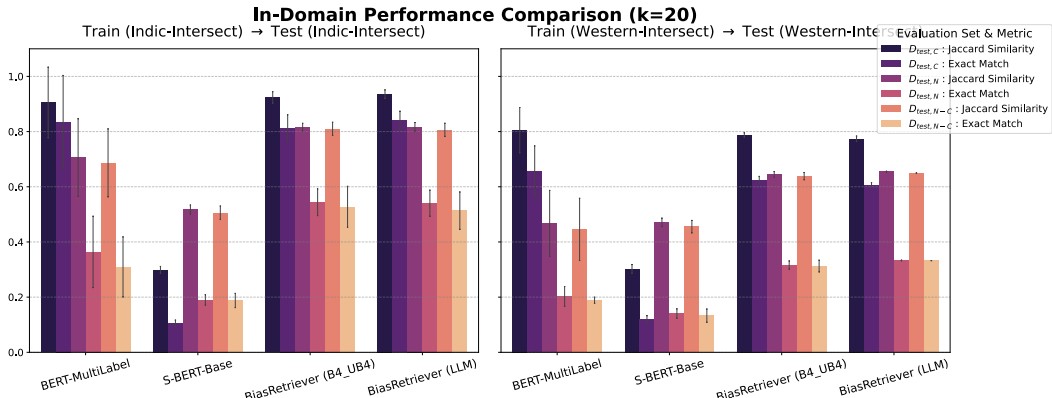

Figure 7: Performance comparison across **In-Domain** scenarios using MiniLM. Our `BiasRetriever` models consistently outperform baselines, especially on the unseen intersectional categories ($D_{test,N-C_i}$) for both corpora, demonstrating superior generalization. The error bars represent standard deviation across three subsets, $C_1, C_2, C_3$.

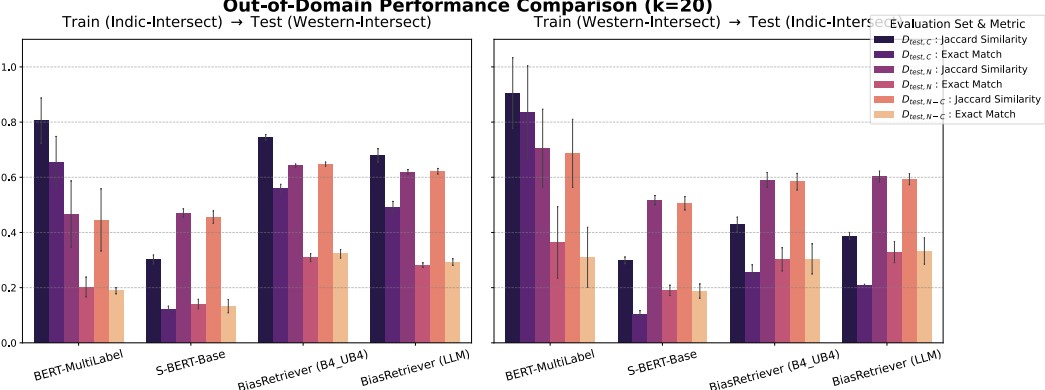

Figure 8: **Cross-domain transfer performance** using MiniLM. Models trained on Indic-Intersect tested on Western-Intersect and vice versa. `BiasRetriever` maintains robustness across domains, unlike classification baselines that degrade sharply.

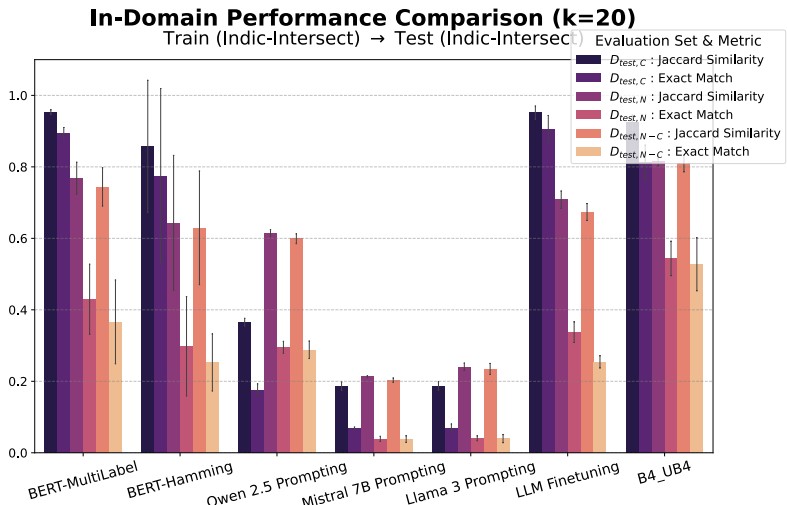

Figure 9: **Intersectional bias detection ablations using LLMs** against MiniLM on *Indic-Intersect*. LLMs perform poorly at bias detection when prompted directly, whereas our proposed BiasRetriever consistently outperforms BERT finetuning and LLM prompting strategies (Figure 21) and achieves stronger detection capabilities.

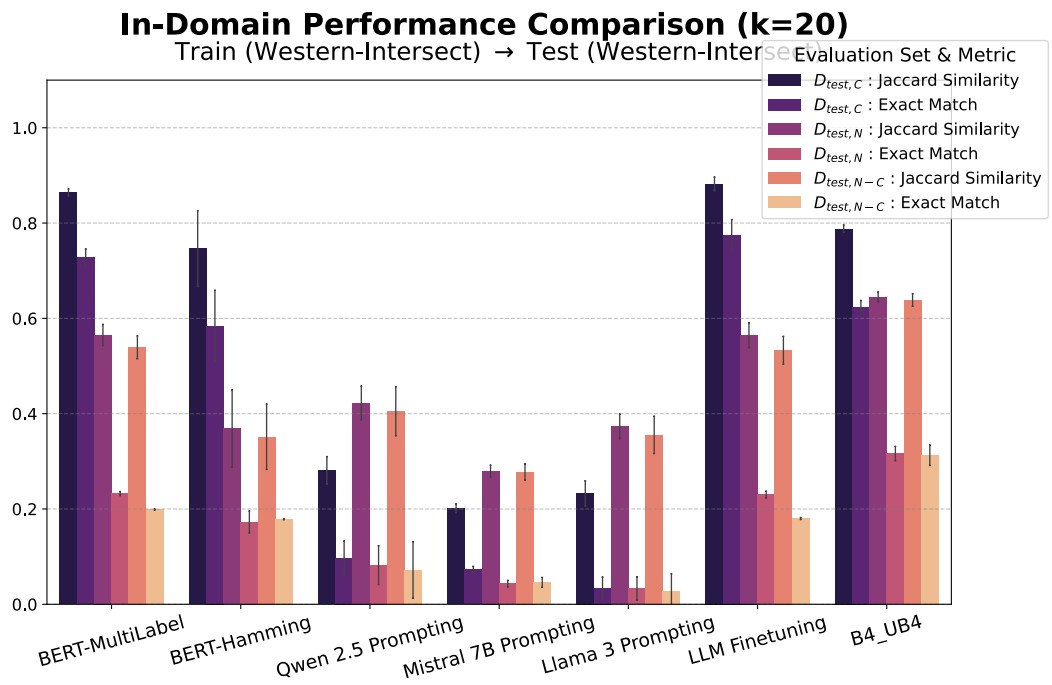

Figure 10: **Intersectional bias detection ablations using LLMs** against MiniLM on *Western-Intersect*. Here also, LLMs perform poorly at bias detection when prompted directly, whereas our proposed BiasRetriever consistently outperforms BERT finetuning and LLM prompting strategies (Figure 21) and achieves stronger detection capabilities.

**Prompt for Biased Sentence Synthesis**

```
You are given two comparative sentences that, together, imply
a biased perspective.  These sentences do not reflect bias
independently, but when paired, they frame one group as less capable,
privileged, or competent.
Your task:  Write a standalone biased sentence that reflects the
underlying bias_type:  {bias_type}.
Do not repeat the two inputs verbatim.
Do not balance or neutralize the view.
Simply generate a plausible, independent sentence that reflects the
combined implication.
Sentence 1:  {modified_eng_sent_more}
Sentence 2:  {modified_eng_sent_less}
Output:
```

**Placeholders:**

- {bias_type}:  Specific bias category being targeted (e.g., "gender", "caste").
- {modified_eng_sent_more}:  First comparative sentence from the IndiBias dataset.
- {modified_eng_sent_less}:  Second comparative sentence from the IndiBias dataset.

Figure 11: The prompt used for synthesizing standalone biased sentences from the comparative pairs in the IndiBias dataset.

A.8    DATA GENERATION PROMPTS

**Prompt for Unbiased Sentence Rewriting**

```
Rewrite the following sentence to remove any {bias_type}-related bias
or stereotype.  The rewrite should:
    • Sound natural, conversational, and realistic.
    • Avoid sounding preachy or corrective.
    • Keep the length close to the original.
    • Make the sentence fair, respectful, and inclusive.
    • You must include the social group in the original {sentence}
      but don't be biased against them.
Sentence:  {sentence}
only return the unbiased sentence.
```

**Placeholders:**

- {bias_type}:  A placeholder for the specific bias category of the source sentence (e.g., "gender", "caste").
- {sentence}:  A placeholder for the original biased sentence that needs to be rewritten.

Figure 12: The prompt used for rewriting biased sentences into neutral, unbiased versions, with placeholder explanations included.

**Prompt for Biased Intersectional Paragraph Generation (Indic-Intersect)**

```
Write a 100-word paragraph that clearly reflects how a {character}
experiences prejudice in everyday life.  The paragraph should be
realistic, coherent, and written in simple, human-like language.
It should be based on a single context, basically the story should
continue in a similar context, and make sure all the biases are
highlighted and faced by the same person.  Use these sentences from
the dataset to shape the character's world:
```

- "{sentence_1}"
- "{sentence_2}"
- ...
- "{sentence_n}"

```
Weave all these ideas naturally into the paragraph, showing how
the intersection of these identities affects how others treat or
perceive the person.  Use clear language and specific examples,
and make sure the prejudices appear explicitly in dialogue or
behavior, not just implied.  Do not use abstract words like "bias"
or "intersectionality," or even bias category words like gender, age,
caste etc.  that would explicitly indicate bias.
```

---

**Explanation of Placeholders:**

- {character}:  A placeholder for the dynamically generated character description based on attributes extracted from the seed sentences (e.g., "young Dalit woman").
- {sentence_1}...{sentence_n}:  Placeholders for the biased sentences sampled from the INDI-REFERENCE corpus, one for each constituent bias in the intersection.

Figure 13: The prompt used for synthesizing biased intersectional paragraphs for the `Indic-Intersect` dataset.

---

**Prompt for Biased Intersectional Paragraph Generation (Western-Intersect)**

```
Write a 100-word paragraph that clearly shows how a {character} faces
unfair treatment in everyday life.  The story should take place in
one realistic setting, with all difficulties experienced by the same
person throughout.  Use these sentences from the dataset to help
shape the character's world:
```

- "{sentence_1}"
- "{sentence_2}"
- ...
- "{sentence_n}"

```
Blend these details naturally into the narrative, making sure
the different aspects of the character's identity interact to
create specific challenges.  Use clear, concrete language and
make the unfair treatment obvious through people's actions or
words|not just implied.  Do not use abstract terms like 'bias' or
'intersectionality,' or mention categories such as gender, age,
caste, or related words.
```

---

**Placeholders:**

- {character}: A placeholder for the dynamically generated character description (e.g., "a disabled Black woman").
- {sentence_1}...{sentence_n}: Placeholders for the biased sentences sampled from the SBIC-REFERENCE corpus, one for each constituent bias in the intersection.

Figure 14: The prompt used for synthesizing biased intersectional paragraphs for the Western-Intersect dataset.

---

**Prompt for Unbiased Intersectional Paragraph Generation (Indic-Intersect)**

**SYSTEM PROMPT:**
Expert narrative writer specializing in positive, inclusive storytelling

**USER PROMPT:**
Create a positive, everyday narrative based on the identities and situations mentioned in these inputs:
- – {sentence_1}
- – {sentence_2}
- – ...
- – {sentence_n}

Create a 100-word paragraph that:

1. Uses the specific identities already mentioned in the sentences naturally
2. Shows the person/people engaging in normal, positive activities
3. Depicts success, achievement, everyday moments, or positive interactions
4. Uses concrete, realistic situations from daily life
5. Maintains a neutral to positive tone throughout
6. Avoids any mention of challenges, problems, discrimination, or negative experiences
7. Focuses on capabilities, achievements, relationships, work, education, or community involvement
8. Let the identities emerge naturally from the context without forcing them

Generated Scenario:

---

**Placeholders:**
- {sentence_1}...{sentence_n}: These are placeholders for the unbiased sentences selected by the semantic coherence-based sampling from the INDI-REFERENCE corpus.

Figure 15: The prompt used for synthesizing unbiased intersectional paragraphs for the Indic-Intersect dataset.

**Prompt for Unbiased Intersectional Paragraph Generation (Western-Intersect)**

```
SYSTEM PROMPT:
Expert narrative writer specializing in positive, inclusive
storytelling

USER PROMPT:
Create a positive, everyday narrative based on the identities and
situations mentioned in these inputs:
      • - {sentence_1}
      • - {sentence_2}
      • - ...
      • - {sentence_n}
Create a 100-word paragraph that:
   1. Uses the specific identities already mentioned in the
      sentences naturally
   2. Shows the person/people engaging in positive activities
   3. Depicts success, achievement, everyday moments, or positive
      interactions
   4. Uses concrete, realistic situations from daily life
   5. Maintains a neutral to positive tone throughout
   6. Avoids any mention of challenges, problems, discrimination,
      or negative experiences
   7. Focuses on capabilities, achievements, relationships, work,
      education, or community involvement
   8. Let the identities emerge naturally from the context without
      forcing them
Generated Scenario:
```
---
```
Placeholders:
      • {sentence_1}...{sentence_n}:  These are placeholders for the
        unbiased sentences selected by the semantic coherence-based
        sampling from the SBIC-REFERENCE corpus.
```

Figure 16: The prompt used for synthesizing unbiased intersectional paragraphs for the `Western-Intersect` dataset.

**Prompt for LLM-Positive**

```
Write a single sentence that shows a clear bias about
'{anchor_bias_clean}'.  Use the following text as inspiration:
"{anchor_text}"
Provide only the biased sentence, with no additional commentary or
explanation.
```
---
```
Explanation of Placeholders:
      • {anchor_bias_clean}:  The intersectional bias category (e.g.,
        "gender and caste").
      • {anchor_text}:  The anchor paragraph text.
```

Figure 17: Prompt for the generating synthetically similar positive samples.

---

**Prompt for LLM-Negative**

```
Write a single, completely neutral and unbiased sentence related to
the context of '{anchor_bias_clean}' but with no bias.
Use the following text as inspiration:  "{anchor_text}"
Provide only the neutral sentence, with no additional commentary or
explanation.
```

---

**Explanation of Placeholders:**

- {anchor_bias_clean}:  The intersectional bias category (e.g.,
  "gender and caste").
- {anchor_text}:  The anchor paragraph text.

---

Figure 18: Prompt for the generating synthetically similar negative samples.

---

**Prompt for Counterfactual Negative**

**SYSTEM PROMPT:**
```
You are a sentence transformer.  Output only the requested
transformed sentence with no explanations, analysis, or additional
text.  Never include reasoning, steps, or commentary.
```

**USER PROMPT:**
```
TASK: Transform this sentence by replacing demographic groups with
different ones from the same category.
SENTENCE: "{positive_text}"
INSTRUCTIONS:
```

- Replace demographic terms:  gender (man→woman), caste
  (brahmin→dalit), age (child→elderly), etc.
- Keep exact same structure and bias
- Handle pronouns correctly (his→her, men→women)
- Output format:  Just the transformed sentence, nothing else

```
TRANSFORMED SENTENCE:
```

---

**Explanation of Placeholders:**

- {positive_text}:  The retrieved positive sentence that will
  be transformed.

---

Figure 19: Prompt for the **Counterfactual Negative** strategy, used to transform a retrieved positive into a hard negative.

**Prompt for Neutral Anchor Paraphrasing**

**System Message:**
You are an expert academic writer specializing in precise paraphrasing.  Output only the paraphrased sentence.

**User Prompt:**
Generate a single sentence that is semantically similar to the following paragraph but phrased differently.  The new sentence must be neutral and unbiased.  Do not add any preamble or explanation.
Paragraph:  "{paragraph}"
Similar Sentence:

---

**Explanation of Placeholders:**

- {paragraph}:  The unbiased anchor paragraph that is paraphrased into a positive sample.

Figure 20: Prompt for the **Neutral Anchor Paraphrasing** strategy, used to paraphrase a neutral anchor into a positive sample.

**Prompt for Intersectional Bias Classification**

**System Message:**
You are an expert at classifying text for intersectional bias. Classify the given paragraph into one of these categories: {categories}.
Instructions:
- Select the single most appropriate label from the list
- When outputting the label, replace "+" with ", " (e.g., "gender+caste" → "gender, caste")
- If the text does not show any bias, respond with "unbiased"
- Do not generate explanations or reasoning, only return the label

**User Prompt:**
Classify this paragraph for intersectional bias:
"{text}"
What is the classification?

---

**Explanation of Placeholders:**

- {text}:  The paragraph to classify.

- {categories}:  Comma-separated label set injected from the chosen dataset; populated from "Indic-Intersect" or "Western-Intersect".

Figure 21: Prompt for Intersectional Bias Classification for both Indic-Intersect or Western-Intersect using Llama-3-8B-Instruct, Mistral-7B-Instruct, Qwen2.5-7B-Instruct

**Prompt for Intersectional Bias Classification**

```
<|im_start|>system
You are an expert at detecting intersectional bias in text.
Analyze the given paragraph and identify any biases present from
these categories: {categories}.
Instructions:
- If multiple biases are present, list them separated by commas
- If no bias is detected, respond with "unbiased"
- Be precise and only identify biases that are clearly present
<|im_end|>
<|im_start|>user
Analyze this paragraph for intersectional bias:
{text}
What biases are present?<|im_end|>
<|im_start|>assistant
{formatted_output}<|im_end|>
```

**Placeholders:**

- {text}: Paragraph to analyze.
- {categories}: Dataset-injected label set (from Indic-Intersect or Western-Intersect).
- {formatted_output}: Gold labels for training (comma-separated or "unbiased").

Figure 22: Prompt used to fine tune Qwen2.5-1.5B-Instruct for intersectional bias classification

**Prompt for Intersectional Bias Classification**

```
<|im_start|>system
You are an expert at detecting intersectional bias in text.
Analyze the given paragraph and identify any biases present from
these categories: {categories}.
Instructions:
- If multiple biases are present, list them separated by commas
- If no bias is detected, respond with "unbiased"
- Be precise and only identify biases that are clearly present
<|im_end|>
<|im_start|>user
Analyze this paragraph for intersectional bias:
{text}
What biases are present?<|im_end|>
<|im_start|>assistant
[Model generates response here during inference]
```

**Placeholders:**

- {text}: Paragraph to analyze.
- {categories}: Dataset-injected label set (from Indic-Intersect or Western-Intersect).

*Note: Assistant response is generated during inference, not predefined.*

Figure 23: Inference Prompt after fine tuning Qwen for intersectional bias classification

