# OpenReview forum: "BEYOND SINGLE-AXIS FAIRNESS: LEARNING TO DETECT INTERSECTIONAL BIASES"
_ICLR.cc/2026/Conference — Submitted to ICLR 2026_

### Official Review · Reviewer_11HG · 2025-10-22

**Soundness:** 1
**Presentation:** 1
**Contribution:** 1
**Rating:** 2
**Confidence:** 4

**Summary:**

This paper considers the problem of intersectional bias evaluation and a dataset is created for this purpose. Moreover, a Biasretriever model is trained for detecting social biases.

**Strengths:**

The problem considered in the paper (intersectional bias) is an important yet less explored one. Creating a datasets for evaluating intersectional biases is a useful contribution.

**Weaknesses:**

- The paper is poorly written, often not mentioning the **reasons** behind a particular design choice and only explaining **what was done**. This makes it difficult to understand (the reader has to predict the reasons that would have made the authors to follow the described process) the paper. For example, the choices of the corpora, why do you need to use the prompt shown in Fig 11 to *synthesise* the bias dataset etc. are not clear from the paper.
- The definition of the social bias types (e.g. do you consider non-binary gender?) are not provided in the paper.
- The human validation is done by only two annotators. No further information is provided regarding those annotators, their social backgrounds.
- The inter-annotator agreement of Kappa 0.91 looks unrealistically high for such a social bias annotation task. Do you have any explanation for this? What guidelines were given to the annotators?
- Typos: A space must be kept between the open bracket of a citation. The double quotes are incorrect in some cases, acknowledgement section etc. are left as is in the template (if you do not need it then do not show it).
- The dataset is not provided even for the review purposes. If the main artefact is the dataset then you should at least provide a subset of it for the reviewers to take a look.
- This paper creates a dataset for evaluating social biases using LLMs. However, no ethics statement is provided in the paper.

**Questions:**

1. Although there is a paragraph about prior work of intersectional bias datasets in the related work section, the paper does not explain the novelty/differences between those datasets and the one created in this work. Could the authors explain this difference please?
2. Can you explain the reasons behind why IndiBias and SBIC were selected as the preferred corpora for creating this dataset?
3. Why did you use all-miniLM-L6-v2 as the sentence encoder? How does social biases in that encoder affect your conclusions?
4. The authors talk about **compositionality** of social biases in the presentation of the hypothesis in Line 326. I was wondering whether you could explain this point further. Intersectional biases could be correlating and might not necessarily be compositional. (e.g. being black and being female might not always mean the negative biases would "add up"?)
5. Related to point 4 above "sexism + racism should embed near single-bias exemplars of both dimensions" also need some explanation. Although the authors claim this to be "intuitive", it is not at all intuitive to this reviewer.

**Details Of Ethics Concerns:**

This paper creates a dataset for evaluating social biases using LLMs. However, no ethics statement is provided in the paper. The dataset itself is also not available for the reviewers to check whether what types of biases are reflected in the dataset. The authors should ensure that the dataset is used only for evaluation purpose and not for training LLMs on it because that could potentially make the trained models socially biased. The authors should demonstrate better awareness of such implications of their work.

---

### Official Review · Reviewer_7AV7 · 2025-10-24

**Soundness:** 2
**Presentation:** 2
**Contribution:** 2
**Rating:** 2
**Confidence:** 4

**Summary:**

The paper introduces BiasRetriever, a contrastively trained retriever designed to detect intersectional biases. The authors build two paragraph-level corpora, Indic-Intersect and Western-Intersect, tailored to distinct sociocultural contexts, and train the retriever to learn a bias-aware embedding space that clusters biased text near canonical stereotypes while pushing it away from unbiased examples. Compared to classifier baselines, BiasRetriever demonstrates notably higher robustness on unseen intersectional categories and cross-domain transfer. The authors argue that this retrieval-based, label-agnostic formulation better captures the compositional structure of intersectional bias and generalizes beyond fixed label sets

**Strengths:**

(1) This paper constructs two new, paragraph-level datasets—Indic-Intersect and Western-Intersect—covering diverse sociocultural contexts.

(2) The proposed method demonstrates strong generalization to unseen intersectional categories and robust cross-domain transfer.

(3) The paper provides clear experimental design and ablation studies (e.g., retrieval depth, triplet generation strategies).

**Weaknesses:**

(1) The most critical limitation lies in the weak and outdated baselines. Comparisons are limited to simple BERT-based classifiers and a frozen S-BERT retriever, without including any state-of-the-art LLMs in zero-shot, few-shot, or supervised fine-tuned settings. This omission leaves it unclear whether BiasRetriever truly advances the state of the art. The task of identifying biased text is not inherently complex and could likely be handled effectively by modern LLMs through in-context learning or supervised fine-tuning, weakening the empirical significance of the proposed gains.

(2) The model’s usefulness in real-world scenarios is unproven; all evaluations are conducted on synthetic LLM-generated datasets, making it uncertain whether the method generalizes beyond these curated corpora.

(3) No code or data released

(4) Both datasets are produced via LLM prompting; without artifact audits, there is a serious risk that performance gains stem from stylistic or lexical cues rather than genuine bias understanding.

(5) Main results are shown only in plots—no full numeric tables, variance, or statistical significance tests are provided.

**Questions:**

(1) Why were no LLM-based baselines (e.g., GPT-4, Llama-3) included, even in zero-shot or few-shot settings? Including stronger models could significantly contextualize the contribution of BiasRetriever.

(2) Can the authors provide evidence that the model generalizes to real-world data (e.g., naturally occurring biased text from social media, forums, or news), rather than just performing well on synthetic, LLM-generated datasets?

(3) Can the authors provide qualitative retrieval examples or embedding visualizations that illustrate how BiasRetriever represents intersectional bias compositionally

---

### Official Review · Reviewer_376K · 2025-10-29

**Soundness:** 2
**Presentation:** 2
**Contribution:** 3
**Rating:** 2
**Confidence:** 3

**Summary:**

The paper aims to provide a framework for the detection and mitigation of intersectional biases, which arise from the intersection of multiple identities, and are distinct from single axis biases. The proposed framework employs a retriever, which is trained  contrastively with new datasets of biased and unbiased paragraphs that were constructed in a multi-step process involving LLMs. Bias predictions are made by aggregating the bias labels of the top-k retrieved sentences in the reference datasets.  The paper provides considerable results for the generalization capabilities across multiple biases and two domains.

**Strengths:**

- Originality: The paper presents a new retriever-based architecture with contrastive learning to solve the task, which can be beneficial in terms of interpretability compared to approaches using labels generated by an autoregressive LLM.

- Quality: Broad testing results of the generalization capabilities and performance of different variants are included, which seem promising.

- Clarity: The paper provides a quite clear explanation of all the steps in the framework.

- Significance: The paper targets a highly important task, as bias remains an unsolved problem especially because of the number of bias dimensions and their interactions.

**Weaknesses:**

- First, the author's claim that their approach "produces higher-quality debiased text than strong fine-tuning and prompting baselines", but there are no results provided on this. All the presented results are related to bias detection not mitigation.

- Second, the paper claims to contribute an approach to "intersectional" bias detection, but it does not convince that the utilized dataset represents unique patterns for intersectional bias. Though the paper states that "For instance, stereotypes about Black women cannot be reduced to the sum of racial and gender stereotypes", they do not succeed operationalizing this. The provided example for "gender+age+physical-appearance" in Table 6 reads like an enumeration of the biases, but it is not clear that there is any interaction leading to a distinct pattern. If the picked example is like this, I am not convinced that the dataset has examples that better represent intersectional biases. Then the task is multi-bias detection instead of a detection of intersectional biases, which is of course still important. One cause in the framework might be that using one distinct sentence per bias cannot adequately represent intersectional bias.  Furthermore, there is an "Intersectional Identity Check" but not an intersectional bias check in the human validation.

**Questions:**

Framing the paper:
The listed contributions do not seem to match the experimental setup and results of the paper. Thus, the claim to "produce higher-quality debiased text" could be removed and the framework could possibly be reframed to "multi-axis bias detection" instead of "intersectional bias detection".

Regarding literature:
There could potentially be more extensive motivation for using the retriever-based architecture, as this is a core feature. There are also existing papers on using contrastive loss for bias detection/mitigation that seem quite relevant but have not been mentioned, e.g.  https://aclanthology.org/2024.findings-naacl.293/  The paper could also be better placed in the bias literature in terms of multi-bias detection not intersectional bias detection.

Other:
- Citations should be checked for updated venues, e.g. "Gender, race, and intersectional bias in resume screening via language model retrieval" is published in ACM.
- The appendix is already well-utilized for providing details, but it would be helpful to also add examples of biased and unbiased reference sentences to the appendix, as well as a list of categories for each bias type.
- line 223: "two" is repeated
- It is also not clear why the wording of the task description in the prompts are so different for the two datasets. E.g. "Weave all these ideas naturally into the paragraph, showing how the intersection of these identities affects how others treat or perceive the person." vs. "Blend these details naturally into the narrative, making sure the different aspects of the character’s identity interact to create specific challenges. " These differences could be explained in the appendix.

---

### Official Review · Reviewer_9R1i · 2025-11-03

**Soundness:** 2
**Presentation:** 2
**Contribution:** 2
**Rating:** 2
**Confidence:** 4

**Summary:**

This paper addresses intersectional bias detection in LLMs through BiasRetriever, a contrastively trained dense retriever that learns bias-aware embeddings. The core argument is that intersectional biases (e.g., race + gender) can be detected by mapping biased text close to constituent single-axis bias examples in embedding space. The authors contribute two paragraph-level datasets (Indic-Intersect and Western-Intersect, 7,404 total paragraphs) and demonstrate up to 10% improvement in Jaccard score over BERT baselines on unseen categories.

Novelty. The core contribution of using retrieval for bias detection is undermined by very recent work: HInter (March 2025) [1] uses mutation-based testing for intersectional bias detection, while BiasAlert (July 2024) [2] already employs retrieval-augmented generation for bias detection. The datasets are smaller than existing benchmarks (Ma et al., 2023 covers 106 groups [3]; contemporary work uses 176K-350K records [4]). The specific triplet mining strategies show methodological rigor but insufficient differentiation from prior contrastive learning approaches.

Significance. The problem is important -- Wilson & Caliskan (2024) show intersectional biases affect 150,000 jobs for Black men alone [5]. Cross-domain transfer capabilities offer practical value. However, the crowded competitive landscape, smaller scale (3-50x smaller than concurrent benchmarks), detection-only scope (no mitigation), and missing comparisons with directly relevant recent work significantly limit impact. This represents incremental progress rather than a breakthrough.

[1] Souani et al. (2025). "HInter: Exposing Hidden Intersectional Bias in Large Language Models." arXiv:2503.11962.

[2] Fan et al. (2024). "BiasAlert: A Plug-and-play Tool for Social Bias Detection in LLMs." EMNLP 2024.

[3] Ma et al. (2023). "Intersectional Stereotypes in Large Language Models: Dataset and Analysis." Findings of EMNLP 2023.

[4] Liu et al. (2024). "Evaluating and Mitigating Social Bias for Large Language Models in Open-ended Settings." arXiv:2412.06134.

[5] Wilson & Caliskan (2024). "Gender, Race, and Intersectional Bias in Resume Screening via Language Model Retrieval." arXiv:2407.20371.

[6] An et al. (2025). "Measuring gender and racial biases in large language models: Intersectional evidence from automated resume evaluation." PNAS Nexus, Volume 4, Issue 3.

[7] Khan et al. (2025). "Investigating Intersectional Bias in Large Language Models using Confidence Disparities in Coreference Resolution." arXiv:2508.07111.

[8] Cantini et al. (2025). "Benchmarking adversarial robustness to bias elicitation in large language models." Machine Learning, Springer.

[9] Zhao & Chang (2020). "LOGAN: Local Group Bias Detection by Clustering." EMNLP 2020.

[10] Narnaware et al. (2025). "SB-Bench: Stereotype Bias Benchmark for Large Multimodal Models." arXiv:2502.08779.

**Strengths:**

- Well-motivated problem: Addresses critical gap in intersectional bias detection with clear real-world implications documented in recent large-scale studies [5,6].
- Rigorous data validation: High inter-annotator agreement (Cohen's κ = 0.91 for sentences, 0.89 for paragraphs) ensures dataset quality.
- Cross-domain generalization: Label-agnostic retrieval formulation enables transfer across cultural contexts where classification baselines fail entirely -- a genuine practical advantage.
- Systematic experimental design: Multiple triplet curation strategies (SR-k4, SR-k10, SR-k4-UN, SR-k4+LLM) with thorough ablations provide methodological insights.
- Cultural diversity: Indic-Intersect dataset addresses underrepresented non-Western contexts, complementing predominantly Western-focused bias research.
- Coherence-based sampling: Semantic coherence optimization for unbiased examples creates challenging negatives that advance beyond simple template-based approaches.

**Weaknesses:**

- Undermined novelty claims: HInter (March 2025) [1] directly addresses intersectional bias detection using automated testing techniques.
- BiasAlert (July 2024) [2] already uses retrieval for bias detection. The paper's differentiation ("we train the retriever itself") is seems insufficient given RAG-based approaches already exist.
- Missing critical baselines: No comparison with HInter [1], BiasAlert [2], WinoIdentity [7], or contemporary LLM-based detection methods [8]. Comparisons limited to BERT fine-tuning and frozen retrievers are ok but seem insufficient.
- Insufficient dataset scale: 7,404 total paragraphs versus 106 groups in Ma et al. (2023) [3], 176K-350K records in contemporaryy benchmarks [4], and 36K resumes in Wilson & Caliskan [5]. Scale concerns limit generalization claims.
- Detection-only contribution: No mitigation strategies while recent work combines detection with debiasing techniques (RLRF, fine-tuning, ensemble methods) [4]. This limits practical deployment value.
- Incomplete related work coverage: LOGAN (2020) [9] proposed clustering-based local bias detection but is not adequately differentiated. - Recent intersectional benchmarks (CLEAR-Bias [8], SB-Bench [10]) are not discussed.
- Weak baseline performance: Figures 9-10 show LLM prompting baselines (Llama 3, Mistral 7B, Qwen 2.5) perform poorly, but this may reflect prompt engineering rather than fundamental approach limitations. No comparison with state-of-the-art prompted LLMs or few-shot methods.
- Limited evaluation metrics: Reliance on Jaccard similarity and exact match may not capture real-world harms. No downstream task evaluation or user studies to validate practical utility.

**Questions:**

- Differentiation from HInter: How does your approach compare quantitatively with HInter [1] on the same datasets? HInter demonstrates that 16.62% of intersectional biases are "hidden" and require explicit testing -- does your retrieval approach detect these cases?
1) BiasAlert comparison: Can you provide direct performance comparison with BiasAlert [2] on your datasets? What specific advantages does training the retriever versus RAG-based detection provide?
2) Scale justification: Contemporary benchmarks use 25-50x more data [4]. What prevents scaling your approach? Have you tested performance with larger datasets?
3) Baseline LLM prompting: Figures 9-10 show poor LLM performance, but recent work [8] shows LLM-as-a-Judge approaches can be effective. Have you tested with stronger prompting strategies (chain-of-thought, few-shot with examples)?
4) Cross-dataset generalization: How does your model perform on existing benchmarks (Ma et al. 2023 [3], BBQ extensions [4])? Can you demonstrate zero-shot transfer to these datasets?
5) Mitigation pathway: What are specific next steps for converting detection capabilities into bias mitigation? Can retrieved bias-matched examples guide debiasing interventions?
6) Computational costs: What are training/inference costs compared to classification baselines and LLM-based approaches? Is retrieval practical for production deployment?

---

### Meta-Review · Area_Chair_pfUt · 2025-12-22

**Summary:**

Reviewers question the work’s novelty, given closely related methods for intersectional bias detection and retrieval-augmented bias analysis. Empirical validation relies solely on small, synthetic LLM-generated datasets with no real-world or human-labeled benchmarks, making the robustness and practical value uncertain. Key baselines, especially modern LLM-based detector, are missing, and the evaluation metrics and reporting lack depth. Claims of "intersectional" modeling are not fully supported by examples or dataset design, which appear more multi-bias than intersectional. Several presentation issues, missing methodological details, limited human validation, absence of dataset release, and lack of an ethics statement further reduce confidence in the contribution.

**Reviewer Concerns:**

No rebuttal provided.

**Reviewer Scores:**

No changes.

---

### Decision · Program_Chairs · 2026-01-26

Reject